**PLOS** NEGLECTED TROPICAL DISEASES

# A qualitative case study of community experiences with Tungiasis in high prevalence villages of Bungoma County, Kenya: *"The whole body aches and the jiggers are torturing me!"*

**Åse Walle Mørkve**[1,2]*, **Jackline Sitienei**[3], **Graziella Van den Bergh**[1,4]

**1** University of Bergen, Bergen, Norway, **2** NLA University College, Bergen, Norway, **3** Moi University, Eldoret, Kenya, **4** Western Norway University of Applied Sciences, Bergen, Norway

* asemor@nla.no

## Abstract

### Introduction

Tungiasis is a painful skin infection caused by a flea called Tunga Penetrans/jiggers, which enters the epidermis of humans and animals. If untreated it may result in bacterial infection, sepsis, necrosis, and disability. In Kenya, it is estimated that 4% of the population suffer from jigger infestation. The aim of this study was to contribute with knowledge about the experiences of those affected, perceived causes and local coping strategies, to improve the control and elimination of this neglected condition.

### Method

A qualitative case study research design was applied involving fieldwork in Bungoma County, a high-prevalence rural area in Western Kenya. Multiple data collection methods were combined: participant observation, home visits, semi- structured in-depth interviews, and group discussions. In total, 48 informants participated, including infected children and adults, teachers and pupils, public health officers, community health workers and NGO volunteers.

### Results

Those infected suffered with multiple penetrating wounds on hands and feet that cause disability, resulting in an incapacity to work and school drop-out. People described feeling stigmatized, and at school pupils preferred not to play with infected classmates. People perceived that the sand flea infestation was caused by poverty and that those affected were not even able to cover their basic needs. They were often living in sandy huts that they shared with their animals, without access to soap and clean water. Moreover, those infected were often viewed as ignorant by the rest of the community. Informants perceived recurrence after treatment as inevitable, resulting in creation of hopelessness. Those infected felt

**Data Availability Statement:** All data can be accessed in OSF. 10.17605/OSF.IO/FQ5KB Contributor name: Åse Walle Mørkve Project name:

"The whole body aches and the jiggers are torturing me!": A qualitative study of community experiences with Tungiasis, in a high prevalence rural area Bungoma, Western Kenya

**Funding:** University of Bergen, Norway, funded the main researcher, Åse Walle Mørkve, with 25 000 NOK during the first fieldwork in 2012. The funder had no role in study design, data collection and analysis, decision to publish, or preparation of the manuscript.

**Competing interests:** The authors have declared that no competing interests exist.

that they were left alone with an irremediable plague. There was confusion about effective approaches regarding prevention and treatment at all levels.

## Conclusion

Tungiasis is a debilitating and neglected ailment, inflicting severe suffering and increasing the circle of poverty. To address fatalist attitudes among those affected, national guidelines need to be implemented, and coordination of public health measures regarding prevention and treatment need to be strengthened. Further research is recommended to enable the control and elimination of this neglected tropical disease

## Author summary

Tungiasis, popularly called jiggers, is a painful and itchy skin infection. Jiggers is a poverty-driven plague and millions of people are at risk. Those infected may suffer mental and physical illness and stigmatization. Despite this, the disease is neglected by the scientific community, the health sector and policy makers. This study contributes to context-sensitive knowledge production necessary to design effective, multi-level interventions, and calls for more implementation research on control efforts and prevention of jiggers' infestation. Qualitative research on the experiences of jigger-affected communities is virtually nonexistent, yet important to identify effective health promotion measures.

## Introduction

Tungiasis is a parasitic skin disease caused by the sand-flea *Tunga penetrans*, popularly called jiggers in English. Jiggers is a poverty-driven plague, and it is one of the emerging neglected diseases in Africa, Latin America, the Caribbean, and the Indian Ocean. It is reported that hundreds of million people in up to 89 countries are at risk of Tungiasis; however, the true prevalence of Tungiasis is unknown [1,2]. It is estimated that in hyperendemic areas, 21–83% of the population may be infected [3,4,5]. In 2022, according to estimations from the Ministry of Health in Kenya, approximately 2 million inhabitants suffer from jiggers' infestation, and 10 million were at risk of being infected [6]. The prevalence of jiggers is highest in poor communities that have lack of water and insufficient sanitation, and where households with non-solid floors share houses with animals [2].

 *T. penetrans* is a mite from the family Tungidae of the species Siphonaptera. The mite is spread by a broad spectrum of animals, with chickens, pigs, dogs, cats, and rats as the principal reservoirs [7]. The flea penetrates the epidermis of a host, human or animal, most likely on the feet but also other body parts that are in contact with the soil. The pregnant female flea burrows into the skin, where it stays for four to six weeks, feeding on blood. Within 24 hours the penetration site gets irritated, and within 2–3 days the site becomes painful. The infection site is also characterized by a visible, albeit almost completely in-buried, flea. As the site becomes irritated, the response is described as an intense itching and scratching will spread hundreds of eggs over several weeks [8]. Without appropriate treatment in endemic communities, people with the infection are at risk of nail loss, formation of ulcers and fissures, inflammation, suppuration [9], chronic lymphedema [10], sepsis and tissue necrosis [11] and gangrene [12]. Those that are not vaccinated against tetanus are at risk of getting it [12], and the infected part of the

body can result in impaired functionality and hinder simple activities such as walking or hand gripping [13].

The early extraction of the flea is the first-line therapy and may prevent further infestation [14] yet a sharp and sterile instrument, skills, time, and appropriate light are needed, which makes treatment options limited in poor areas. Removal is painful, and, if the egg sac ruptures it will cause severe inflammation and re-infestation [10]. Using footwear helps prevent the flea from penetrating [13], and insecticides and fumigation in infested surroundings will lower the risk of infection [15].

This qualitative case study aimed to describe and uncover the lay knowledge that is rooted and interpreted within the context of people's lives and day-to-day experiences [16], close to anthropological understanding of health and illness [17]. According to Kleinman's explanatory model of health and illness, people view their illness in terms of "how it happens, what causes it, how it affects them, and what will make them feel better" [17]. The aim of this study, inspired by Kleinman's theoretical model, is to seek in-depth knowledge about the experiences of those affected, their own explanations of the *causes* of the infection in a high–endemic community, and their quest for treatment. There is scarcity of global research regarding jiggers in highly endemic areas. According to the World Health Organization, tungiasis is listed as a Tropical Neglected Disease (NTD) [18] and tungiasis might hamper the attempt of achieving the Sustainable development Goals (SDG) [19,20]. However, the burden of the disease is still neglected today by the scientific community, the health sector, the pharmaceutical industry and policymakers. Qualitative research is needed [2], addressing the perspectives of those affected, that is essential to guide awareness raising and disease control programs, and for informing policy makers at a local and global level [21,22].

## Methods

### Ethics statement

Ethical approval was given by the Regional Committee for Medical and Health Research Ethics, Western Norway and by the Institutional Research and Ethics Committee (IREC) at Moi University in Eldoret, Kenya before data collection (FAN: IREC 000865). Written informed consent was obtained from all participants before the interviews, and a tape recorder was used when approved by them. When children were interviewed, formal consent was obtained from the parent/guardian. The participants were informed that they could withdraw at any time and that their responses would be deleted. All interview and discussions' material were anonymized.

### Study context

The study was carried out in Bungoma County in the Republic of Kenya in East Africa. Bungoma County is located in the highlands of Western Kenya, on the border to Uganda. Bungoma is the second largest county in Kenya, with a population of almost 1,7 million inhabitants [23]. A Red Cross' Jigger removal program operated in six divisions within Bungoma county, and these were all included in our study: Ndivisi, Chwele, Bumula, Kanduuyi, Nalondo and Amagoro. Typically, each of the divisions in the rural area consisted of 2000–3000 inhabitants. This jiggers removal program has been operational from 2002, with more activities and support during the first 15 years, yet with less outreach periods in later years, such as during the Covid-19 pandemic, and it is still running today (2023), when funding allows it.

The field researcher, the first author participated in the jiggers' removal program conducted in the six divisions (Fig 1). In addition to jiggers' removal and care, the Red Cross staff

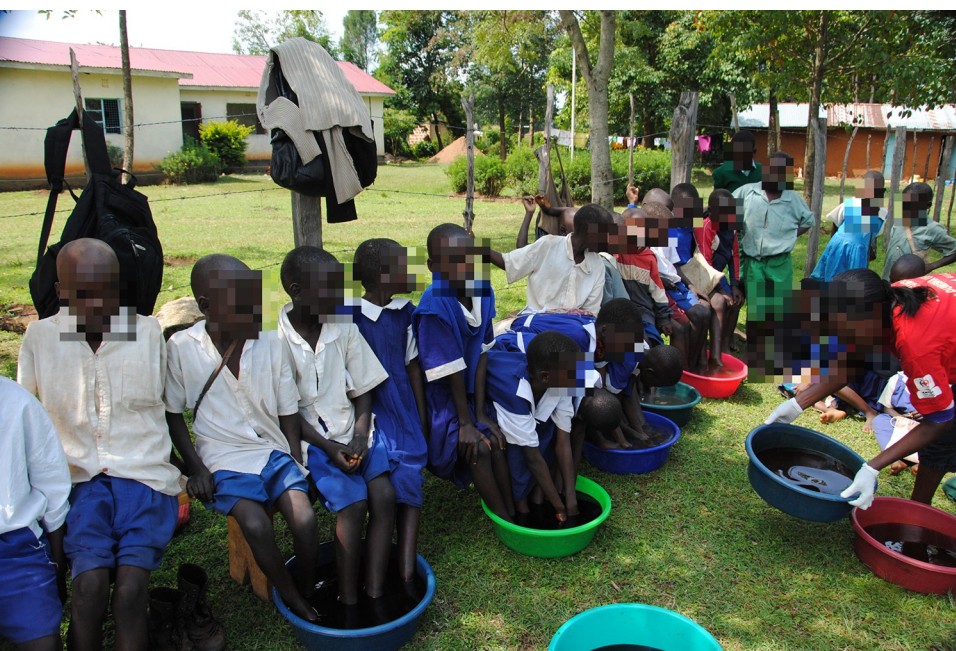

**Fig 1. Jiggers removal program in school, with antiseptic soaking treatment.** Picture taken by first author.

promoted awareness by speaking with those affected, with teachers, health workers and others attending the program. The collaboration with Bungoma Red Cross created the opportunity to explore the context of the plague in-depth, and to understand perceived challenges, needs and resources in communities. The collaboration between the researcher and Bungoma Red Cross was previously established as the researcher was a volunteer in Hordaland Red Cross in Norway, and Bungoma and Hordaland had a twinning program at the time of three fieldwork periods. Two volunteers (who are referred to as 'research assistants') in Bungoma Red Cross were used as research assistants/translators/ guides throughout the fieldwork. The on-going collaboration with patients, volunteers, health workers and other stakeholders ensured a "dialogic epistemology" resulting in multiple perspectives on jiggers' infestation and responses, including beliefs, emotions and motives [16].

## Data collection

To achieve a broad understanding of experiences, qualitative case study design was used during three consecutive periods of fieldwork in Bungoma (one to three months of length between 2012 and 2016). A qualitative case study design involves studying a phenomenon of naturally occurring cases, such as community experiences of jiggers during the jiggers removal program, within its context, such as different villages of Bungoma County. The researcher often spends months in one field, aiming to write in- depth on the phenomenon. Case study design typically involve multiple data collection methods [16]. In this current study data-collection methods included participant observation, home visits, informal talks, semi-structured in-depth interviews, and group discussions, involving 48 informants (Fig 2). In addition, four home visits were conducted after three weeks, to families that had participated in the jiggers removal program (N = 35). These visits included informal talks, where the research assistants took notes, in addition to observation of the participants' homes. Throughout the fieldwork periods a participant approach was conducted, meaning that the main researcher participated as an

| In depth interview (N = 19) | Participants (Total N = 19) | Group discussions (N = 5) | Participants (Total N = 29) |
|---|---|---|---|
| 1) Staff and volunteers of Bungoma Red Cross | 5 | 1) Pupils infected with jiggers (below 18 years) | 5 |
| 2) Staff and volunteers from other NGOs | 4 | 2) Pupils infected with jiggers (below 18 years) | 5 |
| 3) Public Health Officers | 5 | 3) Infected adults (18 – 55 years) | 2 |
| 4) Community Health Workers | 4 | 4) Infected elderly persons (above 55 years) | 5 |
| 5) Infected adult (18 - 55 year) | 1 | 5) Non-infected pupils (below 18 years) | 5 |
| | | 6) Head teachers | 7 |

**Fig 2. Overview of informants.**

assistant in the jiggers' removal program, aiming to get more insight into the program, engaging in dialogue and create trust between the researcher and the participants of the study [16].

The 19 in-depth interviews and 6 group discussions were conducted in a safe environment near the jiggers removal program site; for example, in a local classroom [16]. For the semi-structured in-depth interviews and the group discussions, interview guides were used. The interview guide consisted of ten open-ended questions, and was developed and pilot-tested by the authors and the assistants/guides with the aim to explore experienced causes of the infection, how the infestation affects those suffering it, and their strategies for accessing treatment. However, the interviewees' responses determined the importance of the different topics, and they were encouraged to speak openly and at length [16].

For the discussions, the informants suffering from jiggers were recruited by convenience in a group discussion while naturally assembled and using the same service in a known environment [16]. They were asked during the jiggers' removal sessions, to participate in the discussion. The jiggers removal program was free of charge and was focused on low income communities, implying that the groups of participants were likely to have similar socio-economic background. The research assistants conducted the group discussions in Swahili, whereafter translating and transcribing the interviews. Due to poor participation at one of the jiggers removal programs, one of the intended group discussions was conducted as a semi-structured in-depth interview.

The semi-structured in-depth interviews were conducted in English, by the main researcher, and these were conducted until saturation of information was achieved [16]. The Community Health Workers and volunteers from NGOs were recruited at the Jiggers Removal Program, as they assisted Bungoma Red Cross with the removal program. The local Public Health Officers were other identified key informants who were asked to participate in the study.

### Data analysis

Data were analyzed through a thematic content analysis approach in order to identify the key elements that recurred in all the dataset [16]. In practice, interviews were transferred from a tape recorder into Open Code 3.2 software which organized, structured and coded the data. An example of text divided into meaning units, implying the words, sentences or paragraphs that related to the same central meaning is shown in Fig 3. Meaning units were abstracted and labeled with a code, and codes were then gathered in a category, and finally categories were encompassed into the three following main themes; *Suffering the condition*, *Causal explanations of the jigger plague* and *The creation of hopelessness* (Fig 4). The analysis process was inspired by Kleinman et al (1978) to include the latent content of the text, linking underlying meanings on experienced causes of jiggers, and how it affects those infected, together [16,17].

## Results

Three main themes emerged during analysis (Fig 4); *Suffering the condition*, *Causal explanations of the jigger plague* and *The creation of hopelessness*. Our findings will be illustrated through quotations from participants.

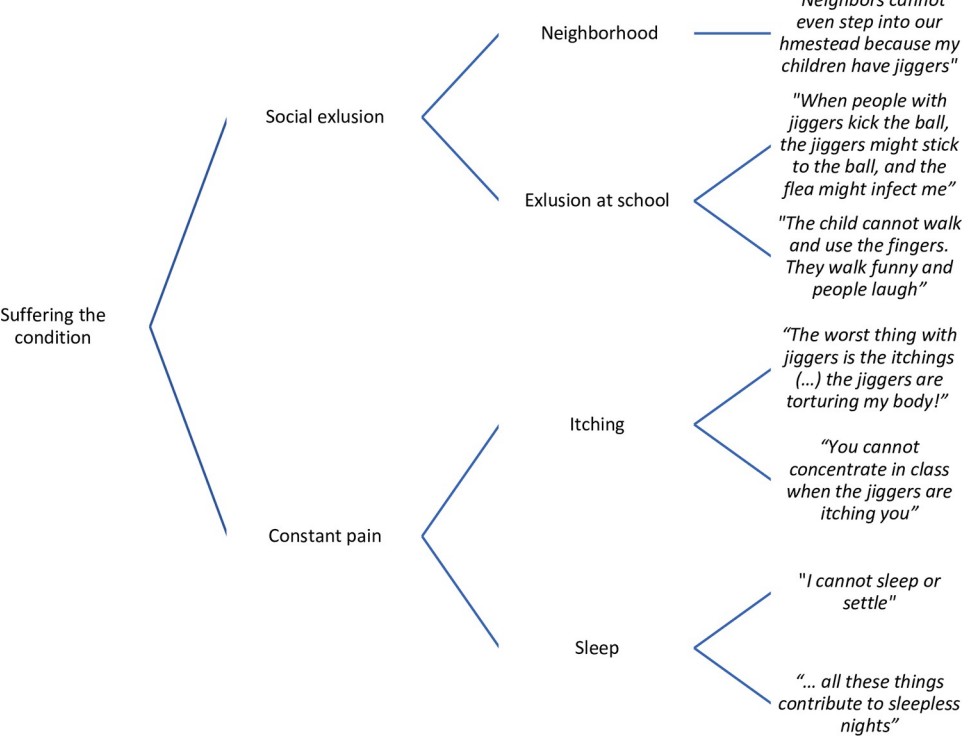

**Fig 3. Data analysis.**

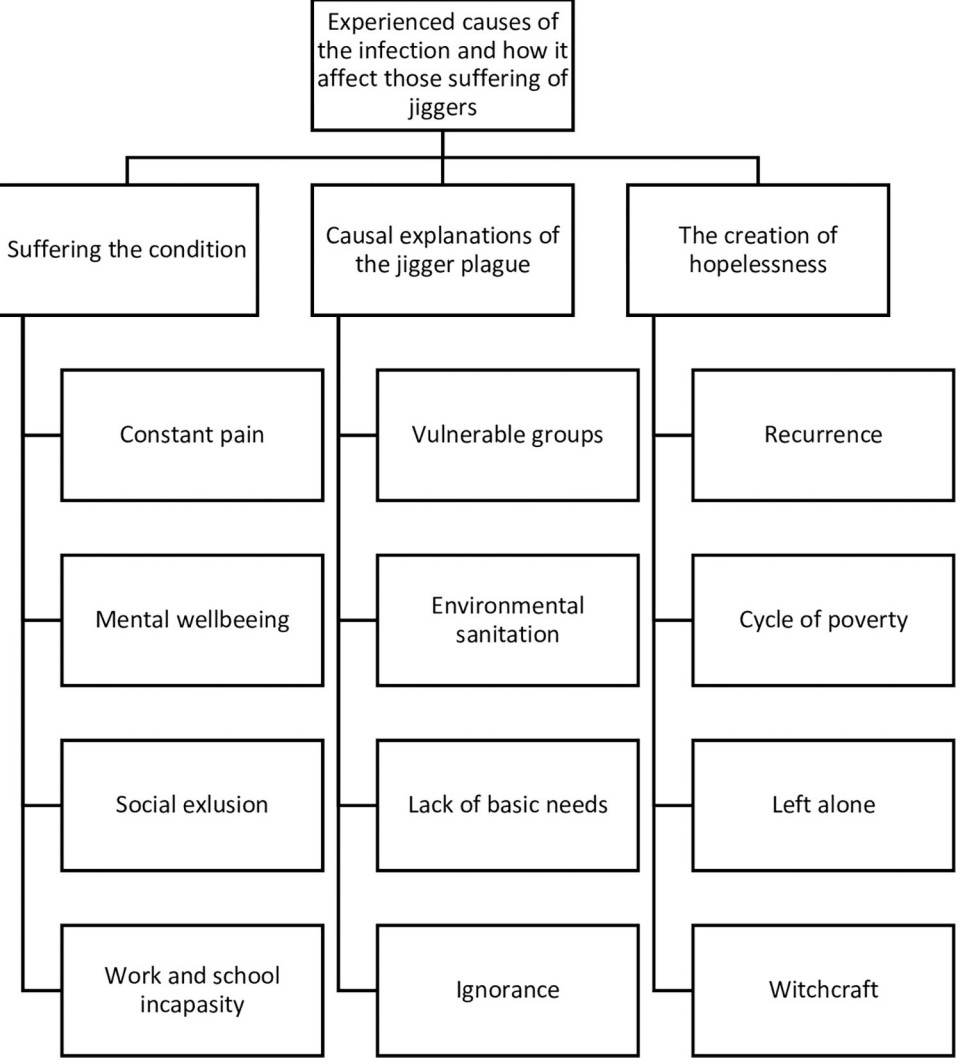

**Fig 4. Framework/ flowchart.**

## Suffering the condition

The constant itching after infection impeded people in to sleep and carry out their daily activities. A woman infected by jiggers explained during one of the group discussions;

> *"The worst thing with jiggers is the itching. I cannot sleep or settle. The whole body aches (. . .) and the jiggers are torturing my body!"*

As repeatedly observed and confirmed, those severely affected could "*not walk, neither work*" because of the pain on the affected parts of the body (Fig 5), and another woman in the group discussion added that:

> *". . .I cannot even hold things properly. . .and not eat with affected fingers".*

Furthermore, people felt stigmatized, neglected, and severe infestation was reported to affected people's mental well-being. The psychological and psycho-social tension caused by

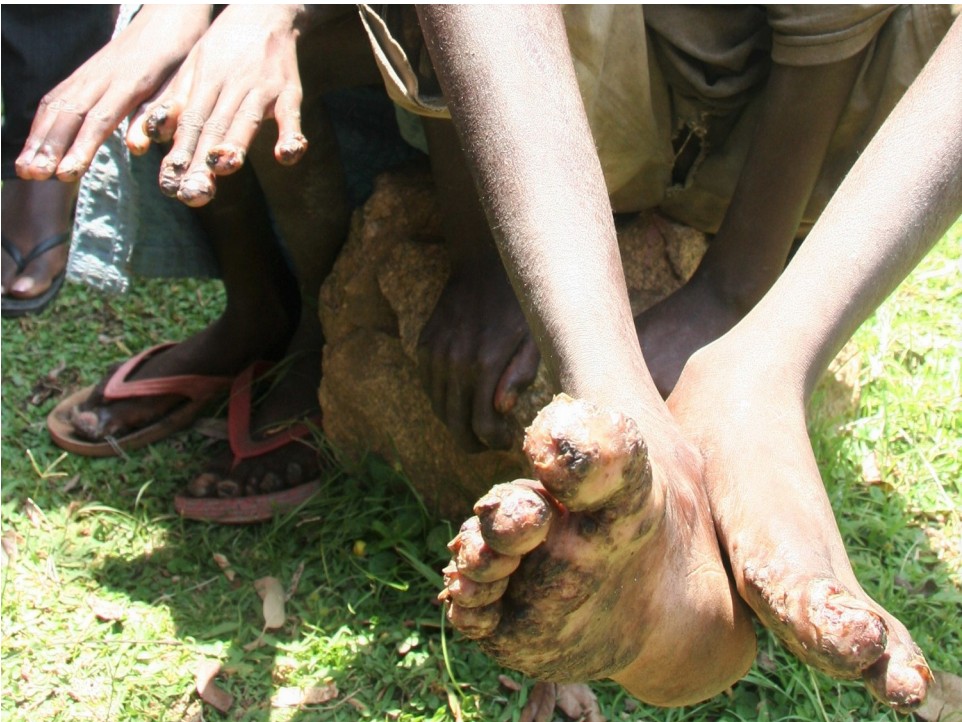

**Fig 5. Children affected by sand-fleas on feets and hands.** Picture taken by first author.

jiggers was a most recurrent theme, and informants agreed that those affected got mentally disturbed by the lack of sleep and the general strain experienced. A woman expressed:

*"I feel that the jiggers attack my brain. I become confused, and I see that others do so too".*

Another man explained during the group discussion that:

*"I feel I get more and more disabled, both physically and mentally".*

This notion was supported by others in the group, and a woman added that:

*". . . all these things contribute to sleepless nights".*

Exploring how those affected were suffering the condition, social exclusion was a recurrent theme in the group discussions. Those affected explained they felt harassed and therefore kept hidden, fearing interactions with others. An infected adult narrated in the group discussion that:

*"Neighbors cannot even step into our homestead because my children have jiggers. They tell their children not to come because they will get infected. . .".*

The fact that those affected were kept isolated, also resulted in poor health seeking behavior. One elderly man explained that: *"I did not want to go to the health center, because I was afraid of being stigmatized".* The four other participants in this group discussion nodded and agreed on his comment. Stigmatization apparently influences health-seeking behavior. Those infected feared being ridiculed by neighbors, at the market or at school, and during jiggers' removal

campaigns. This resulted in people preferring home treatment to keep their condition more of a secret, a phenomenon that was also observed among pupils at school.

As explained by a child with the condition, the jiggers affected his school performance:

*"You cannot concentrate in class when the jiggers are itching you".*

Fear of infection made school-mates afraid of being close to those infected. A non-infected pupil explained that;

*"You must wear shoes when you are with infected people. . . You should also put on protective gloves. . . When people with jiggers kick the ball, the jiggers might stick to the ball, and the flea might infect me".*

During group interviews with teachers, these confirmed that pupils suffering from jiggers were harassed by fellow pupils. One teacher explained that:

*"That shame [is obvious], to see them walking. . .. The child cannot walk and use the fingers* (Fig 6). *They walk funny and people laugh".*

Several of the teachers seemed to have contributed to prejudice amongst those affected, by blaming the individuals who have been infected. In a group interview with four teachers in a rural area, one elaborated that:

*"The main problem is ignorance".*

Alleged ignorance as a causal explanation of the sand flea infestation is among the more frequent locally perceived interpretations of the plague.

## Causal explanations of the jigger plague

Informants in Bungoma widely agreed that children and elderly living in poor conditions were those most infected with jiggers. Both groups are in direct contact with dust and soil, and they often depend on others' support to prevent or to remove the jiggers.

An elderly woman explained to be totally dependent on her grandchildren to remove the fleas from her hands and feet. A Red Cross employee confirmed the vulnerability of children:

*"The adults leave their homes early in the morning and they come back late in the evening. They do not have time to take care of the kids. They strive to provide them with enough food for the day. Children may go for days without taking a bath, using the same clothes and beddings stained with urine. . ."*

Moreover, the inability to fight the jiggers' epidemic due to unmet basic needs emerged as a main explanation. As suggested by several informants, the prime concern was:

"*Food to oneself and the family".*

An employee of Bungoma Red Cross suggested that:

*"If we don't address the poverty issue, we will not be able to address the jiggers' issue".*

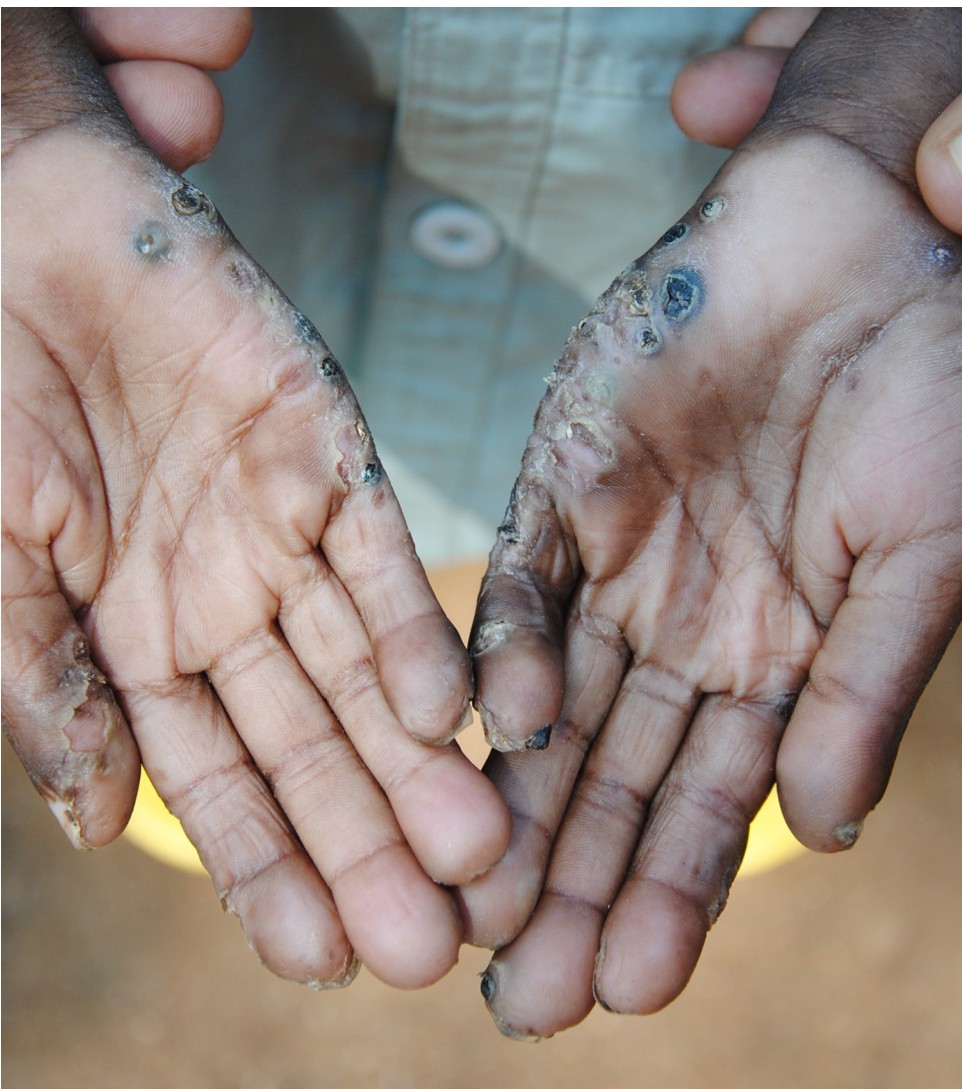

**Fig 6. Pupil that had difficulties writing at school.** Picture taken by first author.

Similarly, a Public Health Officer elaborated that:

> *"If people were not poor, they could have an idea of living in a proper environment. . . I feel that they would not have much problems with jiggers infestation. Then we would be able to eradicate it".*

During fieldwork, and home visits in particular, could the research team observed directly the poor living conditions. All those infected slept on the sandy floor (Fig 7), with direct and frequent contact with animals such as hens, dogs and cows. However, as explained by various informants, having a solid floor and buying beds for the family members were not affordable options. Additionally, keeping animals away from the house and sleeping area was also described as not a feasible option. People and animals lived together in huts, most often as a protection against theft. Several informants explained that they were afraid that their animals would disappear if they did not look after them during the night. When asking in group discussions whether it was common to sleep together with animals, all participants said *"yes"*.

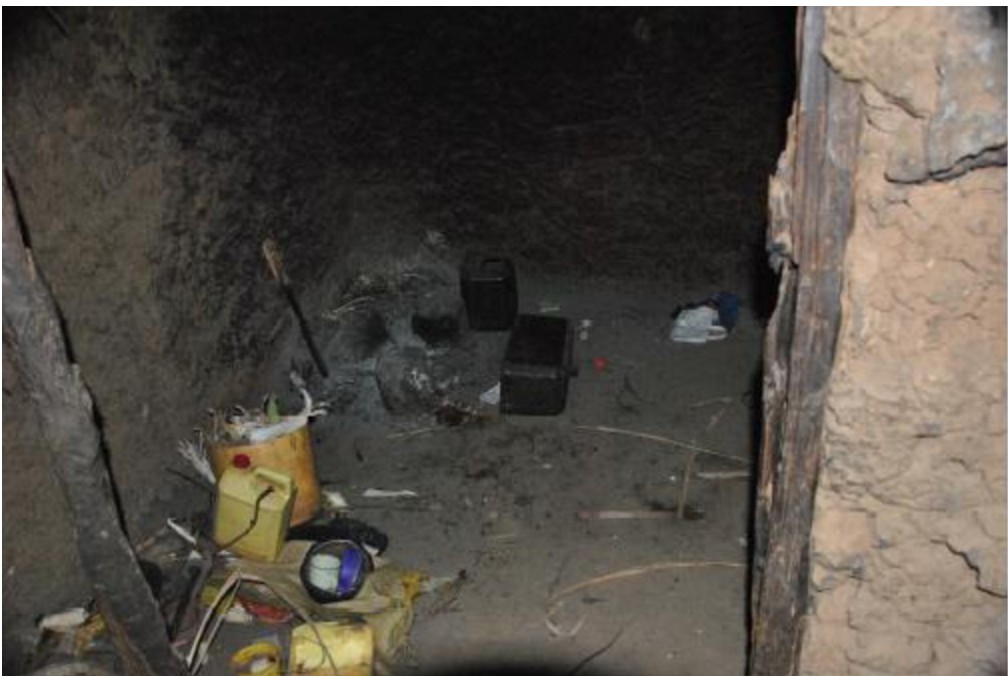

**Fig 7. Sleeping room, where a family of 6 and their animals slept together.** Picture taken by first author.

However, there appeared to be very little knowledge and much confusion about the topic of animals as carriers of the vermin. One infected woman explained:

*"People say that when you go to a place where the goats sleep there are jiggers there".*

However, she was not sure whether this was true, and if other animals than goats implied any threat.

Finally, even though all informants agreed that the underlying cause of the jiggers' scourge was poverty, many health workers also added that even though poor people were most vulnerable, they were also to blame themselves for the issue. For instance, one Public Health Officer elaborated that:

*"The main issue is poverty. . . But also ignorance".*

Another public health officer who seemed aware about the issue of hunger, suggested that public hygiene was expected to be solved by the government:

*"If poor people get money, they use it on food. They do not prioritize hygienic measures. . . We do talk about personal and environmental hygiene. . . But they just wait for it to be bought by the government. . .*

### The creation of hopelessness

Regarding finding a cure or "getting rid of the plague", a sense of hopelessness was commonly described. According to informants, reoccurrence seemed to be difficult to avoid, and virtually all, whether lay-people or professionals, confirmed high reoccurrence or re-infection rates.

Many places, reoccurrence after treatment was perceived as "*inevitable*" and informants expressed that it was somehow "*impossible*" or "*hopeless*" to prevent. As a woman explained, jiggers were «*everywhere*» and; "*. . .we can even get infected when we are sleeping*".

The feeling of disempowerment and the fact that those affected experienced the infestation to occur so suddenly, brought some participants to believe that there were other causes for the problem, such as witchcraft. Some of the people infected explained that witchcraft could be an explanation to why one house could be fully infested, while the neighbor's house did not have any jigger flea in the house, even if they were living in the same conditions. During a group interview, one of the infected pupils almost whispered that "*. . .we believe that maybe we have been bewitched*". Another informant explained, during informal home visits, that the jigger flea came the day after he stole a hen, and he thought that he was bewitched with jiggers due to that.

In Bungoma, virtually all family members in the four households we revisited three weeks after a removal session, were reinfected again. Most of those infected explained that they were relieved after the treatment session, and that they;

> "*At least could sleep without the itching for a short period of time*".

However, within seven days after treatment during the mobile clinics, they started feeling itching again. Another woman explained, supported by others, that

> "*As long as people are living in poor conditions, the jiggers will remain a problem, even if the environment is kept clean*".

A female participant in the jiggers' removal program, who was re-visited, called upon the need for continuous help:

> "*After treatment my health improved, especially the shape of the toes and also the itching stopped. I am now stronger than I used to be, but jiggers have started to re-infest me, and next week I will be weak again. So please, I beg you to continue helping me to get rid of the jiggers. . .*".

As constant re-occurrence seemed to result in perceptions of disempowerment and fatalism, many people seemed to have lost hope about a solution. Some informants also explained that they did not want to go to the dispensaries to get help, because they feared being harassed and that;

> "*. . .We get re-infected again anyways*".

People knew that they should remove the fleas, yet the task is difficult, especially for elders and children, and often resulted in incomplete removal and re-infections. A woman with vermin ridden children explained that:

> "*. . .alone we cannot manage to prevent*".

Some informants added that fatalist attitudes more often occurred when families were separated, where members exposed to jiggers' infection were not capable of paying enough attention to prevent it. A public health officer elaborated that:

*"The breakup family is more vulnerable, where the mother and father do not stay together. And also orphans. . . In some households there are only children, and they do not have a caregiver that can take care of them, that is a problem. And AIDS is a contributing factor."*

Finally, during a home visit, one of those attending the program services begged for continuing receiving help:

*"As long as you are helping, we will be able to prevent it. Alone we cannot manage it. We have stayed in this condition for so long, and we are not able to prevent".*

After this home visit, one of the NGO employees stated that: *"They do indeed sit there and wait for someone coming to help"*, confirming recurring feelings of helplessness and vulnerability in affected communities.

## Discussion

This qualitative study aimed to contribute with perspectives from community-dwelling participants living in high endemic communities, regarding experienced causes of jiggers infestation, how it affects those afflicted by jiggers, and how they seek relief or treatment [16,17], to identify barriers and possibilities for eliminating the plague [17]. Tungiasis is a debilitating and neglected tropical ailment affecting several communities in Kenya and other low-income settings, which is contributes to perpetuating and increasing the circle of poverty [7,24]. Jiggers' infestation is associated with a deleterious effect on quality of life, affecting people's mental and emotional health [25,26], and it is an additional obstacle to achieve the Sustainable Development Goals [20] that will leave vulnerable communities behind.

The issue of jiggers is complex, and in Bungoma there was a wide agreement among all informants that it is not possible to prevent and eradicate jiggers when people live in constant poverty. Previous studies indicate that the prevalence is highest in poor communities and that people suffering from jiggers are less economically productive, which again raises poverty levels as both a cause and effect of infection [14,27]. Poverty in households, in the community, and at national level contributes to perpetuation and re-emergence of such neglected tropical diseases [28]. The need for cost effective control strategies was recognized decades ago, when the Ministry of Health in Kenya launched National Guidelines on Prevention and Control of Jiggers Infestations [29]. However, 5 and 10 years later, the issue of jigger infestation is still neglected in affected counties, whether Bungoma or other parts of the world [2]. Moreover, as did happen with other illnesses and conditions, the Covid-19 epidemic increased the burden and neglect of the sand-flea issue in Kenya [30].

Jiggers infestation affected those suffering the condition in various ways throughout the day and night. Among the affected study participants, disabling pain and itching was extensively reported, resulting in missing sleep. Being contrived by multiple, painful, and disabling penetrations on the hands and feet often resulted in the inability to work and school drop-out, and teachers and pupils alike widely agreed that tungiasis affected school performance. Heavily affected children could not walk to school, they were not able to hold the pen, they felt sick. Research by Ngunjiri in 2015 found that 65,4% of pupils suffering severe jiggers infestation, failed to attend class [19]. Social harassment due to jiggers is a recognized setback among pupils, and it contributes to school drop-out [15,19,25]. Even though there is no empirical evidence in the case of Bungoma County, our observations confirm that children with tungiasis may have high absenteeism at school, and lower performance, as has been observed elsewhere [7,19].

Due to stigma, both pupils and other affected, explained that they preferred to stay at home, and they felt alone trying to combat the jiggers infestation. Trying to keep their condition secret and discrete was also a reason for staying at home, something that was found in other research from Kenya [31]. In Murang'a district in Kenya research by Kimani, Nyangero and Ikamari found that health workers are referring to those affected as illiterate, ignorant and lazy, and other «blaming the poor» attitudes seem to be common. Almost 60% held the opinion that jigger infected persons are lazy [32]. This goes hand-in-hand with jigger sufferers' psychological struggles, which reportedly need more attention [15,27]. Even though jiggers are a huge challenge for several thousand pupils, none of the informants that were asked, including teachers, knew of any school that raised this issue in class. However, the informants emphasized the need for health education, in the community and in schools. They were persuaded that to eradicate jiggers, the level of education had to be improved, not only among those affected but also in the community. This current study indeed supports that teachers, as well as community workers, in affected areas need to increase knowledge on the issue of jiggers, as also teachers stigmatized those affected pupils. Awareness raising among teachers, on pupils suffering various disabilities has been proven effective in other research [33].

Pertaining to environmental preventive measures, health workers, teachers and pupils alike knew that dusty and sandy surroundings were causes of jigger infestation, yet none mentioned potential community-based solutions. Another Kenyan study supports that small-sized and overfilled classrooms made of mud increased the multiplication of the flea [15,34]. Elson et. al. (2019) found that severe tungiasis might be reduced by 50% if sleeping places of children had hardened floors [34].

Furthermore, both domestic and sylvatic animals can be carriers of jiggers, such as dogs, cats, pigs, sheep, goats, horses, cattle, chickens, birds, elephants, monkeys, rats and mice [27]. However, the animals' role as carriers of the vermin was rarely mentioned by those living in high prevalence area, which is also confirmed in other studies from Kenya [9]. During home visits, the research team observed that no proper waste disposal or a pit- latrine were seen. Rats and mice abounded, and thus an important reservoir to control [7,9]. The findings of this study indicate that even though both professionals and lay-people alike seem well informed about jigger infestation and some of its causes, there is a lack of awareness on the fact that animals are important reservoirs. Neither the health workers or teachers nor those affected by jiggers, and not even the veterinarians spontaneously mentioned animals as potential causes of jigger infestation. Elson et al (2017) elaborate that any control campaign should include an element of community education regarding the role of animal reservoirs [35]. Still, the habit of protecting one's own live stock from theft by keeping them inside may be difficult to change in a short term, and animal and environmental disinfestation may need to be offered at a higher level of governance.

Regarding treatment and eradication of the infection, flea-repenetration is commonly reported, and an important cause of the resulting hopelessness and fatalism among those affected. The feeling of disempowerment and the fact that the infestation occurred so suddenly also brought some to believe that there were other causes for the problem, such as witchcraft. However, this was clearly a topic that was difficult discussing for the informants, and most of them because shy of speaking when this theme occurred during the discussions. Witchcraft believes might increase stigma and hinder those affected from seeking help, as they do not believe there is anything to do with their condition [32].

The high infestation-rate also led to hopelessness. A Brazilian study for instance, reported that children got on average 15 new fleas per week [10]. In an average family with three to four children, this amounts to 6 to 8 fleas that the caregivers need to remove every day. In our study, home visits were conducted three weeks after the jigger removal campaign. The informants reported an increased quality of life after the removal program, however after three

weeks virtually all people were re-infected with the sand flea, and the informants called for continuous help. The issue of frequent reoccurrence [10] favors the use of primary preventive measures, such as using shoes and awareness raising, to fight the jiggers epidemic rather than secondary prevention and the use of drugs to kill the penetrated flea [36]. According to the national guidelines, the traditional use of materials like slime mixed in ash to regularly smear floors and walls of dwelling houses removes breeding places or repels jigger fleas, as does the traditional method of dust-suppression by regular wetting of dusty floors. The use of plant extracts as repellants e.g. neem, aloe vera and coconut oil as topical applications has also been shown to be effective and environmentally friendly jigger repellents which also induce considerable remission in clinical pathology [29]. A study by Thielecke et.al. (2013) also found promising results using the plant–based repellent Zanzarin twice a day to reduce the attack rate [37]. However, virtually all informants in the study called for clear guidelines and strategies on preventive measures, as confusion on this topic also somehow led to fatalism. None of the professionals in Bungoma had access to brochures or written information on prevention and treatment of jiggers, thus illustrating the paucity of knowledge about how to deal with the plague. An eclectic variety of methods (such as using sharp needles) were applied to prevent and treat jiggers, by those affected as well as by the NGO workers, documenting the need for evidence-based recommendations. Before our study, public information about the jigger situation in Kenya was scanty and fragmented [32]. In 2014, The Division of Environmental Health at the Ministry of Health in Kenya published guidelines with the aim to control and eventually eliminate jigger infestations, through strengthening capacity for sustainable prevention and control at the community, county and national levels; to guide the use of pesticides for control in infested households, schools and other institutions; to guide the activities of various actors, whereof health workers; to advocate for surveillance mechanisms for jigger prevalence and evaluation of control programs and finally to stimulate research on the issue [29]. However, despite the presence of an NGO investing in controlling jiggers in Bungoma county, effective, standardized approaches to prevention, control and treatment still need to be implemented, and community empowerment boosted. By 2023, Red Cross continues to engage in humanitarian work on jiggers when they have funding, yet according to a local Public Health Officer in Bungoma, they still have no jiggers' situation analysis data since "they do not have a tool to collect these" (personal communication). As in other counties and countries, sand flea infestation is still a serious public health problem that is poorly recognized, and data are scarce [21,25,38]. To sum up, documenting the extent of the problem, economic means and governmental support for environmental action and implementing safe preventive measures, and awareness raising, in the community, with teachers, public health officers, community-, county- and national leaders is necessary.

## Study limitations

In this participatory study, the research assistants were recruited from Bungoma Red Cross and helped to recruit relevant informants, to translate and during data collection. The fact that they had limited experience in conducting qualitative interviews, that in turn needed translation and transcription from Swahili or Bukusu to English, might have resulted in some degree of bias in the data. However, the length of the fieldwork, the immersion in the communities while participating in local jigger clinics and the continuous, open, and critical dialogue and counter checking with the research assistants and with the leaders of Bungoma Red Cross, before, during and after the fieldwork helped outweigh these potential biases. Also, multiple data collection methods were used including thick descriptions, for a better understanding of local concepts and of the context.

## Conclusion and recommendations

This qualitative case study explored local perceptions of being afflicted by and living in jigger infested communities; how people explained the scourge, and how they coped with it in a rural Kenyan high-prevalence county, with the aim to better understand how to fight the disease. The flea affects those infected in terms of physical, psychological, and social well-being, in particular children and older people, who often depend on others to prevent and physically remove the flea. Being a very stigmatizing condition, it impedes many to seek help. Moreover, the high degree of contamination often occurred together with poverty and unsatisfied basic needs. An important and pervasive perception was about the unavoidable reoccurrence of infestation, expressing fatalism in affected communities, alongside lack of awareness about animals being flea reservoirs. Several informants exhibited hopelessness and lamented that no matter what they did, the flea would multiply and continue infecting people in the community.

Addressing jiggers in high-prevalence communities needs a holistic and multi-sectorial One Health approach [38] that is sensitive to people's perceptions, beliefs and hope. It demands coordinated public health measures including both treatment and preventive measures, such as improved sanitation and awareness about animals as reservoirs. Jiggers as a condition of the poor attracts little research; however, it is critical to recognize, reduce and control this NTDs [28]. In line with the National Guidelines on Jigger Control [29], and in line with the Sustainable Development Goals [20], following recommendations seem particularly relevant:

1. Foster collaboration among county and national key stakeholders (Ministries and departments of Health, Education, Science, Water, Agriculture and academic and research institutions, industries, communities, organizations, victims etc,) and the private and non-governmental sector in strengthening jigger prevention, control and eventual elimination.

2. Identify and map areas and communities mostly affected in terms of vulnerability to jiggers and conduct comprehensive baseline surveys and qualitative action research on effective and preventive interventions, while involving people at the grassroots. This to target the infested households or schools for maximum impact while using the Ministry's Community Health Strategy and community health volunteers.

3. Providing material and economic support when sensitizing and mobilizing communities on the significance of personal hygiene and other environmentally friendly jigger prevention methods, while setting up sustainable measures towards ensuring good hygienic practices and behavior change. This includes long term water-, sanitation- and hygiene (WASH) interventions, housing improvement and poverty alleviation, along with providing biological and chemical repellents to victims and those at risk for regular use, including neem solution which is environmentally friendly and non-toxic, for the treatment of house floors and outdoor resting areas.

4. Continue with antiseptic solutions soaking and treatment of afflicted people's feet and other infested body parts to prevent secondary infections and consequences such as disability and death, and follow-up of the victims; continue with outreach and treatment camps and finally administration of Tetanus vaccine to jigger victims and those at risk.

5. Support the development of user- and environment friendly and affordable methods of jigger prevention, control and treatment, including capacity building and manuals for use by health practitioners and other actors.

## Acknowledgments

We want to express our gratitude to all those who shared their painful experiences, to health workers, and staff and volunteers of Bungoma Red Cross who shared their knowledge about this stigmatized condition and public health problem. To Lucas Matias Jeno, Catherine Monika Schwinger and the research group for "intercultural studies, inclusion and social justice" at NLA University College, thank you for input and support.

## Author Contributions

**Conceptualization:** Åse Walle Mørkve, Graziella Van den Bergh.

**Data curation:** Åse Walle Mørkve.

**Formal analysis:** Åse Walle Mørkve, Graziella Van den Bergh.

**Funding acquisition:** Graziella Van den Bergh.

**Investigation:** Åse Walle Mørkve, Jackline Sitienei.

**Methodology:** Åse Walle Mørkve, Graziella Van den Bergh.

**Project administration:** Åse Walle Mørkve, Graziella Van den Bergh.

**Resources:** Åse Walle Mørkve.

**Supervision:** Jackline Sitienei, Graziella Van den Bergh.

**Validation:** Åse Walle Mørkve, Graziella Van den Bergh.

**Visualization:** Åse Walle Mørkve.

**Writing – original draft:** Åse Walle Mørkve, Graziella Van den Bergh.

**Writing – review & editing:** Åse Walle Mørkve, Graziella Van den Bergh.

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
