## [Editor Report · Decision Letter 0]

7 Oct 2022

Dear Mrs Mørkve,

Thank you very much for submitting your manuscript "“The whole body aches and the jiggers are torturing me!”: A qualitative study of community experiences with Tungiasis in a high prevalence rural area in, Western Kenya" for consideration at PLOS Neglected Tropical Diseases. As with all papers reviewed by the journal, your manuscript was reviewed by members of the editorial board and by several independent reviewers. In light of the reviews (below this email), we would like to invite the resubmission of a significantly-revised version that takes into account the reviewers' comments. 

This qualitative study is not conducted according to SRQR checklist (https://journals.lww.com/academicmedicine/fulltext/2014/09000/Standards_for_Reporting_Qualitative_Research__A.21.aspx). Please refer to the reference and draft the manuscript according to the checklist. The findings should not be merged with the discussion section. This study requires tables and figures to describe the major results, rather than just narration of the findings.

We cannot make any decision about publication until we have seen the revised manuscript and your response to the reviewers' comments. Your revised manuscript is also likely to be sent to reviewers for further evaluation.

Sincerely,

Tauqeer Hussain Mallhi, Ph.D

Academic Editor

Victoria Brookes

Section Editor

Dear Authors, thank you for submitting the manuscript to PLOS NTD. I reviewed this manuscript initially before peer-review and found that manuscript needs your attention. This qualitative study is not conducted according to SRQR checklist (https://journals.lww.com/academicmedicine/fulltext/2014/09000/Standards_for_Reporting_Qualitative_Research__A.21.aspx). Please refer to the reference and draft the manuscript according to the checklist. The findings should not be merged with the discussion section. This study requires tables and figures to describe the major results, rather than just narration of the findings.
---

## [Decision Letter · Decision Letter 1]

14 Feb 2023

Dear Mrs Mørkve,

Thank you very much for submitting your manuscript "“The whole body aches and the jiggers are torturing me!”: A qualitative study of community experiences with Tungiasis in a high prevalence rural area in, Western Kenya" for consideration at PLOS Neglected Tropical Diseases. As with all papers reviewed by the journal, your manuscript was reviewed by members of the editorial board and by several independent reviewers. In light of the reviews (below this email), we would like to invite the resubmission of a significantly-revised version that takes into account the reviewers' comments. 

We cannot make any decision about publication until we have seen the revised manuscript and your response to the reviewers' comments. Your revised manuscript is also likely to be sent to reviewers for further evaluation.

Sincerely,

Tauqeer Hussain Mallhi, Ph.D

Academic Editor

Victoria Brookes

Section Editor

Dear Authors, thank you for submitting in Plos NTD. Your manuscript has been assessed by relevant experts from the field. They found the manuscript interesting but raised some concerns in methodology and interpretation of results. It is requested to please consider the comments of reviewers.

In addition to the comments of the reviewers, I would like to suggest following changes;

1. Please consider the modification of the title emphasizing a clear objective of the study, with study design and location. I would suggest to avoid initial statement in the title of this manuscript.

2. Please provide clear information on how the interview guide was prepared.

3. Please provide the details of number of samples included in this study, along with total number of themes extracted from qualitative analysis at the start of results.

4. The first sentence of the limitation section is not clear, rather than to mention first and second author, the authors should just describe the major limitations, if one limitation is addressed by any of the authors, it would not be considered as limitation.

Important: I recommend that authors consider having their work edited by a professional service or a native English speaker.

Reviewer's Responses to Questions

**Key Review Criteria Required for Acceptance?**

**Methods**

-Are the objectives of the study clearly articulated with a clear testable hypothesis stated?

-Is the study design appropriate to address the stated objectives?

-Is the population clearly described and appropriate for the hypothesis being tested?

-Is the sample size sufficient to ensure adequate power to address the hypothesis being tested?

-Were correct statistical analysis used to support conclusions?

-Are there concerns about ethical or regulatory requirements being met?

Reviewer #1: -Are the objectives of the study clearly articulated with a clear testable hypothesis stated? - YES

-Is the study design appropriate to address the stated objectives? - YES

-Is the population clearly described and appropriate for the hypothesis being tested? - YES

-Is the sample size sufficient to ensure adequate power to address the hypothesis being tested? - YES

-Were correct statistical analysis used to support conclusions? - N/A

-Are there concerns about ethical or regulatory requirements being met? - NO

Reviewer #2: Although the study design has been well described, the aim of the survey could be formulated a little more clearly (especially in the abstract). It is also not entirely clear according to which criteria the study participants were selected.

**Results**

-Does the analysis presented match the analysis plan?

-Are the results clearly and completely presented?

-Are the figures (Tables, Images) of sufficient quality for clarity?

Reviewer #1: -Does the analysis presented match the analysis plan? - Partly

-Are the results clearly and completely presented? - YES

-Are the figures (Tables, Images) of sufficient quality for clarity? - NO

Reviewer #2: The description of the results is good and corresponds to the evaluation criteria

**Conclusions**

-Are the conclusions supported by the data presented?

-Are the limitations of analysis clearly described?

-Do the authors discuss how these data can be helpful to advance our understanding of the topic under study?

-Is public health relevance addressed?

Reviewer #1: -Are the conclusions supported by the data presented? - YES

-Are the limitations of analysis clearly described? - YES

-Do the authors discuss how these data can be helpful to advance our understanding of the topic under study? - YES

-Is public health relevance addressed? - YES

Reviewer #2: The summary is also successful and meets the criteria! Perhaps the recommendations for action in practice could be formulated more clearly (as a list?) in order to give the topic more weight.

**Editorial and Data Presentation Modifications?**

Reviewer #1: See Summary and General Comments.

Reviewer #2: "minor revisons"

**Summary and General Comments**

Reviewer #1: This work is extremely relevant, inducing the reader to a deep reflection on a little explored disease, but which is a serious public health problem in vulnerable populations such as the one this study addressed.

This reviewer has a few considerations to make:

• There is the absence of an appropriate legend in table 1, in the topics defined by the asterisks, which are inserted in the table itself. As a suggestion, the information should appear as a legend and not as an integral part in one more line of the table itself.

• The variables used for data collection can be described more clearly, with the construction of a table or a flowchart, demonstrating the structure of the topics used, including the "key words" captured by the recording system.

• As a suggestion, in addition to clinical illustrative images, due to the severe environmental context described in the results, if available, it would be essential to insert images of the households and the surroundings, emphasizing the sanitary problems that permeate the occurrence and recurrence of tungiasis in this population studied.

• It is very clear that the authors draw attention to the relevance of the social context and poverty that involve the population affected by tungiasis; however, while a world with more equity and social justice does not emerge as the solution to almost all the problems that afflict humanity, it is necessary to think of strategies that can collaborate to the improvement of the health conditions of these people. 

• In this aspect, this reviewer missed the authors' mention of therapeutic strategies that may be promising, such as the use of occlusive agents, especially manufactured dimethicone-based products, as mentioned in a systematic review. In another study, Schwalfenberg et al. (2004) demonstrated in a small case series that the use of coconut and jojoba oil in natura may be useful in avoiding reinfection [1, 2].

Congratulations to the authors on this beautiful work!

References:

1. Tardin Martins AC, de Brito AR, Kurizky PS, et al. The efficacy of topical, oral and surgical interventions for the treatment of tungiasis: A systematic review of the literature. PLoS Negl Trop Dis. 2021;15(8):e0009722. Published 2021 Aug 20. doi:10.1371/journal.pntd.0009722

2. Schwalfenberg S, Witt LH, Kehr JD, Feldmeier H, Heukelbach J 2004. Prevention of tungiasis using a biological repellent: a small case series. Ann Trop Med Parasitol 98: 89-94.

Reviewer #2: The topic is interesting and significant. The manuscript is complete and, all in all, very well written and fully meets the evaluation criteria!

PLOS authors have the option to publish the peer review history of their article (what does this mean?). If published, this will include your full peer review and any attached files.

Reviewer #1: No

Reviewer #2: No
---

## [Decision Letter · Decision Letter 2]

12 Apr 2023

Dear Mrs Mørkve,

We are pleased to inform you that your manuscript 'A qualitative case study of community experiences with Tungiasis in high prevalence villages of Bungoma County, Kenya: “The whole body aches and the jiggers are torturing me!”' has been provisionally accepted for publication in PLOS Neglected Tropical Diseases.

Best regards,

Tauqeer Hussain Mallhi, Ph.D

Academic Editor

Victoria Brookes

Section Editor

Dear Authors,

Thank you for revising the manuscript.

Reviewer's Responses to Questions

**Key Review Criteria Required for Acceptance?**

**Methods**

-Are the objectives of the study clearly articulated with a clear testable hypothesis stated?

-Is the study design appropriate to address the stated objectives?

-Is the population clearly described and appropriate for the hypothesis being tested?

-Is the sample size sufficient to ensure adequate power to address the hypothesis being tested?

-Were correct statistical analysis used to support conclusions?

-Are there concerns about ethical or regulatory requirements being met?

Reviewer #1: -Are the objectives of the study clearly articulated with a clear testable hypothesis stated? - YES

-Is the study design appropriate to address the stated objectives? - YES

-Is the population clearly described and appropriate for the hypothesis being tested? - YES

-Is the sample size sufficient to ensure adequate power to address the hypothesis being tested? - YES

-Were correct statistical analysis used to support conclusions? - YES

-Are there concerns about ethical or regulatory requirements being met? - YES

**Results**

-Does the analysis presented match the analysis plan?

-Are the results clearly and completely presented?

-Are the figures (Tables, Images) of sufficient quality for clarity?

Reviewer #1: -Does the analysis presented match the analysis plan? - YES

-Are the results clearly and completely presented? - YES

-Are the figures (Tables, Images) of sufficient quality for clarity? - YES

**Conclusions**

-Are the conclusions supported by the data presented?

-Are the limitations of analysis clearly described?

-Do the authors discuss how these data can be helpful to advance our understanding of the topic under study?

-Is public health relevance addressed?

Reviewer #1: -Are the conclusions supported by the data presented? - YES

-Are the limitations of analysis clearly described? - YES

-Do the authors discuss how these data can be helpful to advance our understanding of the topic under study? - YES

-Is public health relevance addressed? - YES

**Editorial and Data Presentation Modifications?**

Reviewer #1: (No Response)

**Summary and General Comments**

Reviewer #1: Congratulations on the new version of the manuscript!

PLOS authors have the option to publish the peer review history of their article (what does this mean?). If published, this will include your full peer review and any attached files.

Reviewer #1: No

<quillbot-extension-portal></quillbot-extension-portal>

---

## [Editor Report · Acceptance letter]

21 Apr 2023

Dear Mrs Mørkve,

We are delighted to inform you that your manuscript, "A qualitative case study of community experiences with Tungiasis in high prevalence villages of Bungoma County, Kenya: *“The whole body aches and the jiggers are torturing me!”*," has been formally accepted for publication in PLOS Neglected Tropical Diseases.

Best regards,

Shaden Kamhawi

co-Editor-in-Chief

Paul Brindley

co-Editor-in-Chief
